# Classification of Fluorescently Labelled Maize Kernels Using Convolutional Neural Networks

**DOI:** 10.3390/s23052840

**Published:** 2023-03-06

**Authors:** Zilong Wang, Ben Guan, Wenbo Tang, Suowei Wu, Xuejie Ma, Hao Niu, Xiangyuan Wan, Yong Zang

**Affiliations:** 1School of Mechanical Engineering, University of Science and Technology Beijing, Beijing 100083, China; 2Beijing Zhong Zhi International Institute of Agricultural Biosciences, Beijing 101200, China; 3Shunde Innovation School, University of Science and Technology Beijing, Beijing 528300, China; 4School of Chemistry and Biological Engineering, University of Science and Technology Beijing, Beijing 100083, China

**Keywords:** deep learning, fluorescent protein gene, machine vision, maize kernel, sorting system

## Abstract

Accurate real-time classification of fluorescently labelled maize kernels is important for the industrial application of its advanced breeding techniques. Therefore, it is necessary to develop a real-time classification device and recognition algorithm for fluorescently labelled maize kernels. In this study, a machine vision (MV) system capable of identifying fluorescent maize kernels in real time was designed using a fluorescent protein excitation light source and a filter to achieve optimal detection. A high-precision method for identifying fluorescent maize kernels based on a YOLOv5s convolutional neural network (CNN) was developed. The kernel sorting effects of the improved YOLOv5s model, as well as other YOLO models, were analysed and compared. The results show that using a yellow LED light as an excitation light source combined with an industrial camera filter with a central wavelength of 645 nm achieves the best recognition effect for fluorescent maize kernels. Using the improved YOLOv5s algorithm can increase the recognition accuracy of fluorescent maize kernels to 96%. This study provides a feasible technical solution for the high-precision, real-time classification of fluorescent maize kernels and has universal technical value for the efficient identification and classification of various fluorescently labelled plant seeds.

## 1. Introduction

Maize—the most cultivated food crop worldwide—is widely used in food processing and as a major ingredient in animal feed. With the growing annual demand for corn, the demand for maize breeding and seed production technology has increased. The utilisation of maize heterosis—hybrids produced using hybrid seed production technology—can lead to a 30–40% increase in maize yield, among other factors [1]. Traditional hybrid maize kernel production relies on manual or mechanical destemming, which is costly and has relatively low hybrid seed purity [2]. Using a nuclear sterile line as the female parent for hybrid seed production significantly reduces costs while markedly increasing yields and hybrid seed purity. The difficulties resulting from the mass propagation and preservation of maize sterile lines can be overcome by using a multi-control sterility (MCS) system. In this system, the fertility of a sterile line is restored using a fertility gene and the transgenic pollen is aborted by the pollen self-elimination gene. The MCS system uses red fluorescent protein (RFP) genes as kernel selection markers to create female sterile lines of quality maize varieties using hybridization, backcrossing, and other techniques, thereby enabling the effective propagation and production of maize [3]. The transgenic maintainer and non-transgenic sterile lines cultivated using the maize MCS line are used to produce the next generation of non-transgenic sterile line seeds and for maize hybrid breeding and seed production, thereby ensuring efficient use of sterile maize kernels [4]. To effectively distinguish the maintainer line from the sterile line in the maize MCS system, mCherry [5] or DsRed [6] fluorescent protein genes are implanted into the kernels of the maintainer line for labelling and sorting. Therefore, the automatic high-precision sorting of fluorescent maize kernels is vital for the industrial application of the MCS system. 

A combination of machine vision (MV) and automatic sorting devices is considered to be the best technique for sorting kernels and is being widely utilized in various aspects of engineering technology [7,8,9,10], including seed-industry engineering [11,12,13,14]. In terms of image acquisition devices, a large number of scholars use industrial cameras, such as charge-coupled devices (CCDs) and CMOS, to efficiently acquire various image information from kernels [15,16,17,18,19]. Wang et al. [20] used four high-resolution colour CMOS industrial cameras that were distributed evenly around the objects being measured at intervals of 90 degrees, together with four 12 V LED white light sources that were placed between adjacent cameras to provide uniform illumination for multi-angle photography of maize ears and kernels. In the field of kernel image-processing technology, many recognition and diagnosis methods have been proposed based on a pipeline approach that includes image segmentation, feature extraction, and pattern recognition [20,21,22,23,24,25,26,27,28,29,30]. Wang et al. [28] removed impurities of various sizes from among the kernels by comparing the aspect ratio and projected area of the maximum bounding rectangle with those of the maize kernels. Because aflatoxin-infected maize emits fluorescence under a 365 nm ultraviolet light, Zhang [30] proposed an HSV model to identify normal, mildly diseased, and severely diseased maize kernels based on thresholds of the H and V variables, achieving a detection rate of 91%. Convolutional neural networks (CNNs), which can more efficiently screen out the effective parameter information and make the recognition and classification results more accurate by using large amounts of data, have been widely applied to the recognition and classification of plant species in recent years to further improve the generalization ability of seed-kernel recognition models [30,31,32,33,34,35]. To facilitate the advancement of cutting-edge corn breeding technology, Altunta et al. [34] used a transfer learning method with CNNs to automatically distinguish between haploid and diploid maize kernels.

Unlike the subjects of the previously published studies, fluorescently labelled maize kernels require an excitation light source of a specific wavelength to excite fluorescent protein and generate an emission spectrum [22]. The image acquisition system must quickly recognize the dim fluorescent images of kernels in an obscure environment during the sorting process; thus, the image attributes are not particularly essential. Furthermore, different fluorescent proteins and growth settings result in variable luminescence positions and intensities of fluorescent corn seeds, necessitating a seed-detection algorithm with strong model generalization abilities. However, in separating fluorescent corn seeds, all the above approaches encounter technical challenges.

In this study, deep learning techniques are used to achieve the high-precision real-time classification of fluorescent corn seeds. An image acquisition system for fluorescent kernels is constructed and the ideal combination of excitation light source and industrial camera filter wavelength is determined by examining the RGB and HSV colour characteristic features of the fluorescent corn seeds. The effectiveness of various CNN algorithms in recognizing fluorescent kernels is investigated. Finally, the main sorting device prototype is developed and the sorting of fluorescent corn seeds is tested in real time. The results show that the high-precision real-time sorting of fluorescent corn seeds can be effectively realized by using the sorting device designed in this study and the improved YOLOV5 model.

## 2. Materials and Methods

### 2.1. Maize Kernel Samples

We collected experimental samples of fluorescent maize kernels (MCS0701-2 and MCS0703-9) from the Biology and Agriculture Research Centre of the University of Science and Technology in Beijing, China [3]. The samples included three types of kernels in two categories, which were cultivated using the MCS system. The first category was the fertile maintainer line of fluorescent seeds, which carries the fluorescent protein mCherry (MCS0701-2) or DsRed (MCS0703-9), as shown in Figure 1a and Figure 1b, respectively. The second category was the non-fluorescent seeds of sterile lines that do not carry fluorescent proteins, as shown in Figure 1c. In total, we collected 3000 samples, comprising 1000 seeds carrying mCherry, 1000 seeds carrying DsRed, and 1000 seeds without a fluorescent protein. The classification is presented in Table 1. We aimed to differentiate fluorescent fertile kernels from non-fluorescent sterile kernels.

### 2.2. Machine Vision System

#### 2.2.1. Image Acquisition Hardware

The image acquisition device (Figure 2) was a CCD industrial camera with a target size of 1/2.7 inch and a pixel size of 5 million pixels. A USB 3.0 high-speed transmission interface and a wide-angle C-mounted lens with a focal length of 5 mm and a target surface size of 1/1.8 inch were part of the MV system. The system also allowed for height adjustments through the camera mount. The calculated area that needed to be captured by the camera was 600 × 400 mm^2^ (refer to the design requirements in Section 2.3). Hence, the position of the camera was fixed at a distance of 300 mm from the detection surface so that the collected images clearly captured the luminescence of the maize kernels. The area of the collection zone was fixed. The machine-vision system used three LED ring light sources, one of which was nested in the centre of the CCD industrial camera and positioned higher than the camera lens as the main excitation light source for the seed fluorescence. The other two LED ring light sources were placed on the left and right sides of the main excitation light source so that the light intensity in the seed identification area was consistent. To observe the differences between the fluorescent and non-fluorescent seeds, a closed black box was designed to provide a dark image-acquisition environment. To reduce the reflection of the LED light in the dark environment, black matte screens were affixed to the surfaces of the dark box.

#### 2.2.2. Excitation Light Source and Filter Selection

As previously stated, the correct combination of excitation light source and camera filter was required to obtain high-quality images of fluorescent kernels. The spectra of mCherry [5] and DsRed [6] fluorescent proteins are shown in Figure 3. As shown, the optimal excitation wavelength of the two fluorescent proteins was between 540 and 590 nm. For the LED light source, the wavelength of the green light was between 555 and 570 nm and the wavelength of the yellow light was approximately 585 nm [19], which covered the optimal excitation wavelengths of the two fluorescent proteins. Therefore, the green and yellow LED light sources were selected to excite the fluorescent proteins in the maize kernels. The fluorescent protein spectra showed that the maximum emission wavelengths of the two fluorescent proteins ranged from 555 nm to 650 nm. Regarding the filter selection, according to the spectra presented in Figure 3, the maximum emission wavelengths of the two fluorescent proteins ranged from 555 to 650 nm. Because the wavelength of the excitation light source was <600 nm, the selection of a narrow-band filter below 600 nm would have caused the light from the excitation light source to enter the image acquisition system and affect the image quality. Therefore, narrow-band filters with central wavelengths of 610, 630, 645, and 650 nm were selected. However, it was observed that maize kernels with the DsRed fluorescent gene could not be observed through the 610 and 630 nm filters and the white endosperm of the seeds reflected the green light when a green LED light source was used, which affected the recognition accuracy. Finally, two feasible fluorescence excitation configurations, as shown in Table 2, were determined.

To select the optimal configuration from the two listed in Table 2, the RGB colour space and HSV colour moment of typical maize kernel images produced by the two configurations were analysed. It is necessary to pre-process and segment the RGB images of corn seeds captured by CCD industrial cameras. First, the RGB images were grey-scaled to improve the image processing speed. Thereafter, the median filter algorithm was used to smooth the kernel debris in the images and the noise in the image transmission. A Laplace differential algorithm was used to sharpen the images and enhance the edges of the images with small scales. Finally, attached kernels were separated using morphological processing and the edges of separated sub-regions were extracted using morphological image-correction techniques. A watershed segmentation algorithm was used to separate images of multiple kernels into multiple single-kernel images (Figure 4a).

Figure 4b,c and Table 3 show the RGB and HSV colour feature parameters of the extracted kernels following the pre-processing and segmentation of the kernel images. As shown, both configurations allowed the mCherry-1 and Non-3 seeds to be distinguished using RGB colour spaces. However, the two configurations exhibited more noticeable differences in the H and V parameters of the HSV colour moment when DsRed-2 and Non-3 kernels were distinguished. This resulted in the image features of these two seeds becoming more distinguishable, which, in turn, facilitated their sorting and recognition using a deep learning algorithm. The Yellow+645 configuration was chosen as the excitation light source and filter configuration for sorting the fluorescent and non-fluorescent maize kernels.

### 2.3. Maize Kernel Sorting System

#### 2.3.1. Structural Design

A comprehensive seed-sorting system was designed based on the MV system designed in Section 2.2. The structure of the system is shown in Figure 5a,b with the three classification channels for maize kernels. In each sorting channel, the maize kernels first fall one by one at a certain frequency from the motorised seed-drop device ① onto the conveyor belt ②. Then, the conveyor belt ② drives the seeds into the black box of the MV system ④ at a certain speed guided by the channel partitions ③. The MV system ④ captures the images in real time and transmits them to the master computer ⑧ for real-time identification, and the sorting device ⑨ uses the identification information to place the kernels into their corresponding channels. Finally, the same types of seeds are dropped into their corresponding seed boxes ⑦ by the blanking device ⑥ for sorting.

In the sorting device ⑤, each sorting channel is divided into two channels by the channel partition ③ to receive fluorescent maize kernels and non-fluorescent maize kernels, respectively. A steering gear (⑤-1) drives the paddle (⑤-2) to swing according to commands to close the relevant channel, causing the seeds to enter the corresponding channel according to their type.

#### 2.3.2. Control System Design

Figure 5b shows the sorting device control system. A laptop with an Ubuntu 18.04 operating system, an Intel i7-9500h processor, and a GTX 1650 graphics processing unit was used as the master computer ⑧ for real-time image recognition. PyTorch 1.7.0 (meta Company, Menlo Park, California, USA) was used as the framework for deep learning (DL). The host computer ⑧ and Raspberry Pi 4B+ ⑨ were connected to the same router ⑪ using a network cable to ensure that they were both in the same local area network (with a connection established through the IP address). The actions of the three steering motors (⑤-2) in the sorting device ⑤ were directly controlled by the Raspberry Pi 4B+ ⑨ through the control board ⑩.

During sorting, the master computer ⑧ collected real-time information regarding the running speed of the scoop drop device ① and the conveyor belt ② to calculate the positions of the kernels on the conveyor belt. The master computer ⑧ received real-time images of the kernels in each sorting channel from the MV system ④ and performed fluorescence feature recognition and position tracking. Finally, the master computer ⑧ transmitted the motion commands obtained by the Raspberry Pi 4B+ ⑨ to control the steering gears (⑤-1) and sort the seeds.

### 2.4. Image Acquisition and Dataset Production

The spatial resolution of the images taken by the industrial camera was 1920 × 800 pixels but it was adjusted to 640 × 640 pixels in the network model to reduce the computation time and enable more efficient processing. A total of 5000 images of randomly positioned kernels were taken using the MV system, and among these, 4000 were randomly selected as the training set and the remaining 1000 were used as the validation set. Labellmg and Make Sense (Wroclaw, Poland) software were used to label the object location and category information in each image. Then, the XML file tag was generated using Labellmg for the training of the YOLOv3 model and Make Sense generated a text file (txt format) as the tag for the training of the YOLOv5s model.

### 2.5. Kernel Sorting Based on YOLOv5s Network

#### 2.5.1. YOLOv5s Network

Redmon et al. proposed a regression-based object recognition algorithm called YOLO and developed three versions of the algorithm: YOLOv1, YOLOv2, and YOLOv3 [36,37,38]. Compared with region-based convolutional neural networks [39], YOLO models have a considerably improved detection speed while maintaining accuracy [35]. Compared with other YOLO algorithms, YOLOv1 is inadequate for detecting dense and small targets and often results in large positioning errors. YOLOv2 requires pre-training and presents difficulties in data migration, whereas the YOLOv3 model is more complex and performs poorly in detecting medium and large targets. Additionally, the YOLOv4 model has a large body and high hardware requirements. In contrast, the YOLOv5s [40] algorithm has improved efficiency and accuracy in recognition and the fastest processing speed for a single-frame image at 7 ms. As a result, the YOLOv5s is the optimal DL algorithm for the real-time recognition of fluorescent kernels.

The architecture of the YOLOv5s network is shown in Figure 6. In the input part of the model, the Mosaic data augmentation technique combined images that had been randomly cropped, scaled, and arranged to improve the training speed and small object detection. The adaptive display algorithm was used to assign an initial anchor value for generating a prediction boundary in the initial frame, and the deviation from the actual frame was calculated to update the network parameters in reverse. Focus was used to slice the image in the backbone network. In this study, the input image size of the YOLOv5s network model was 640 × 640 × 3. After the slicing operation, a value was taken from every other pixel in the picture to output a 320 × 320 × 12 feature map, which not only completed the downsampling operation but also preserved the image information. A feature pyramid network (FPN) and a path aggregation network (PAN) were used in the neck network of the YOLOv5s network. The FPN employed a top-down approach and used top-level strong semantic features with low-level weak semantic features to achieve a strong fusion of the semantic information. The PAN structure combined low-level strong positioning information with high-level weak positioning information to strongly fuse the semantic features with the positioning information.

#### 2.5.2. Enhancing the YOLOv5s Kernel Detection Network

To address the issues of the low recognition accuracy of the fluorescent maize kernels and the occurrence of redundant prediction frames caused by weak fluorescent images, the YOLOv5s algorithm was improved. Image data augmentation was conducted via gamma transformation [39], and the introduction of a convolutional block attention module (CBAM) [41] enhanced the target localisation, improving the loss function and increasing the accuracy of the prediction frame.

Data augmentation via gamma transformation

Once the kernel images were obtained, they were enhanced with gamma transformation. The principle of gamma transformation is to perform a nonlinear operation on the grey value of the input image, with an exponential relationship between the input and output grey levels. The equation is as follows:(1)$$s=cr^{\gamma }$$
where s is the output pixel grey value, *r* is the input pixel grey value, *c* is a constant, and *γ* is the gamma indicator. When *γ* > 1, the grey-scale histogram is compressed, causing it to move into low grey-scale values and making the image darker. When *γ* < 1, the grey-scale histogram is stretched and moves into high grey-scale values, causing the image to become brighter. The grey-scale input–output relationship is shown in Figure 7a.

Images of DsRed-2 and Non-3 are often difficult to distinguish (refer to Section 2.2.2), as some areas of the images blend with the dark background. Gamma transformation can be used to enhance the contrast between the kernel and the image background. The results of the gamma transformation are shown in Figure 7. When 0 < γ < 0.5, although image brightness and contrast increased significantly, the difference between DsRed-2 and Non-3 was reduced. When 0.5 ≤ γ < 1, the kernel detection against the dark background was enhanced, particularly at γ = 0.7, when the kernel area could be clearly distinguished from the dark background and the differentiation of DsRed-2 and Non-3 was optimal. Therefore, a γ value of 0.7 was selected for image enhancement.

Addition of the CBAM module

We added a CBAM to the YOLOv5s convolutional network to more accurately locate and track the positions of kernels in images and realize real-time detection. The attention module was based on the human visual system and could locate important feature information while suppressing other unnecessary information. The CBAM consisted of a channel attention module and a spatial attention module. The channel attention module identified the feature target and the spatial attention module determined the location information of the extracted target. The outputs of the two attention modules were added and normalized with the sigmoid function to obtain the corresponding weight coefficients, as shown in Figure 8. Feature weight 1 of the input feature was obtained from the channel attention module and was multiplied by the input feature to obtain the intermediate feature. The intermediate feature was then passed through the spatial attention module to obtain the corresponding weight 2, which was multiplied by the intermediate feature to obtain the final feature. The use of max-pooling in the module can retain maximum information and the CBAM can be used as a plug-and-play lightweight module for pre-existing CNN architectures to make model training more efficient without affecting the computational speed.

Improving predicted bounding box accuracy with CIOU_Loss

Kernels were randomly positioned on the conveyor belt; thus, the relative positions of the predicted bounding boxes and target bounding boxes were considerably different. When the intersection over union (IOU) values of multiple predicted bounding boxes are the same as the target bounding boxes, there are redundant predicted bounding boxes, which can lead to biased predictions. The generalized IOU (GIOU_Loss) loss function in the classic YOLOv5s algorithm can only solve scenarios in which the predicted and target bounding boxes do not overlap. When different predicted bounding boxes overlap with a target bounding box with the same region area (IOU value), it is impossible to distinguish their relative positions in the target bounding box, resulting in prediction errors of the kernel positions.

The complete *IOU* loss (*CIOU_Loss*) function is shown in Equations (2)–(4), where *b* and $b^{gt}$ are the centre points of the predicted and target bounding boxes, respectively; ρ is the Euclidean distance between the centre points; *c* represents the diagonal distance of the prediction box from the smallest rectangle of the target bounding box; *v* is the consistency penalty term used to measure the aspect ratios of the two boxes; and *a* is the weight parameter penalty term. It can be seen that CIOU_Loss introduced a series of penalty terms to GIOU_Loss. This enables it to solve scenarios in which the overlap area of multiple predicted bounding boxes is the same as a target bounding box by calculating the centre point distance between the predicted bounding boxes and the ground-truth bounding box. Furthermore, when the centre point of a predicted bounding box is the same as that of the target bounding box, the optimal predicted box can be determined by detecting the aspect ratios of the two boxes. Thus, the positions of the kernels can be localised more accurately and the predicted bounding box redundancy is greatly reduced. Therefore, we used CIOU_Loss as the loss function for the improved YOLOv5s model.
(2)$$CIOU=\frac{\rho^{2}\left(b,b^{gt}\right)}{c^{2}}+\alpha \upsilon$$
(3)$$v=\frac{4}{\pi^{2}}{\left(\arctan\frac{w^{gt}}{h^{gt}}-\arctan\frac{w}{h}\right)}^{2}$$
(4)$$\alpha =\frac{v}{\left(1-IOU\right)+v}$$
(5)$$LOSS_{CIOU}=1-IOU+\frac{\rho^{2}\left(b,b^{gt}\right)}{c^{2}}+\alpha$$

#### 2.5.3. Model Parameters and Assessment Indicators

The size of the input images in the YOLO model was 640 × 640 × 3. The number of images required for a single iteration batch was set at 16. Batch normalization was performed when updating the weights. The learning rate was initially 0.01, and after 300 iterations, the learning rate was 0.0032. The number of iterations was 1600. Momentum was set at 0.843; the Mosaic data were used for augmentation; and the gamma, hue, saturation, and lightness were set at 0.7, 0.0138, 0.664, and 0.464, respectively. During sorting, only fluorescent kernels needed to be identified; therefore, the number of categories *N* of the test samples was set at 1.

To evaluate the performance of the various models, the quantitative metrics of precision, recall, accuracy, F1 score, and average precision (*AP*) can be calculated using the following equations [37]. The detection categories true positives (*TP*) (the number correctly detected as fluorescent kernels), true negatives (*TN*) (the number correctly detected as non-fluorescent kernels), false positives (*FP*) (the number incorrectly detected as fluorescent kernels), and false negatives (*FN*) (the number incorrectly detected as non-fluorescent kernels) were used. The *AP* can be computed by the area under the precision–recall curve. Additionally, the inference time of the model can be extracted to evaluate the computational efficiency.
(6)$$\mathrm{Pr}ecision=\frac{TP}{TP+FP}$$
(7)Recall=TPTP+FN
(8)$$Accuracy=\frac{TP+TN}{TP+TN+FP+FN}$$
(9)F1=2Precision×RecallPrecision+Recall
(10)$$AP=\int_{0}^{1}P\left(R\right)dR$$

## 3. Results and Discussion

### 3.1. Comparison of Model Results

The proposed model was compared with the YOLOv3, YOLOv5s, and Faster R-CNN models in order to illustrate the superiority of the improved YOLOv5s model proposed in this study.

The losses of the YOLOv3, YOLOv5, and the improved YOLOv5s models during training are shown in Figure 9. The convergence speed of the YOLOv3 model was slower and the final loss value was approximately 2, which was significantly higher than the other models. The convergence speed of the YOLOv5s algorithm was faster than that of the improved YOLOv5s algorithm but its final loss value was comparable at approximately 1. This indicated that the YOLOv5s and the improved YOLOv5s algorithms had faster convergence speeds and better convergence results.

To compare the fluorescent kernel detection results, the Fast R-CNN, YOLOv3, YOLOv5s, and improved YOLOv5s models were used to detect one randomly acquired image of 12 fluorescent maize seeds for identification. Figure 10 shows the results of each of these models. The detection results of the YOLOv3 model were poor and multiple detections were missed. The Fast R-CNN model achieved considerably better results than the YOLOv3 model, but some detections were missed. The YOLOv5s model achieved considerably better results than the YOLOv3 model but there were some redundant prediction boxes. The improved YOLOv5s algorithm accurately detected all the fluorescent kernels with no redundant prediction boxes; therefore, it was considered the optimal algorithm for real-time sorting systems.

To quantitatively compare the recognition abilities of the different models for fluorescent maize kernels, the P–R curves (which are shown in Figure 11), AP50 values (IOU threshold of 50%), F1 scores, and inference times of the four models were compared using image data from the model validation set, as shown in Table 4 and Figure 12. The detection efficiency of the Fast R-CNN model was significantly lower than that of the other three algorithms; therefore, it cannot be used in a real-time sorting system. The YOLOv3 model achieved the worst scores for all indicators, indicating that it was unable to achieve accurate, real-time sorting of fluorescent maize kernels. Although the inference time of the improved YOLOv5s model was slightly longer than that of the YOLOv5s model, its AP50 values and F1 scores were significantly better (both exceeding 95%), indicating that the improved YOLOv5s model achieved accurate, real-time sorting of fluorescent maize kernels in the sorting system prototype.

### 3.2. Experiments and Discussion

To verify the effectiveness of the fluorescent kernel sorting system supported by a deep learning model, 500 fluorescent kernel samples (250 mCherry-1 and 250 DsRed-2) and 500 Non-3 kernel samples were mixed and placed in the sorting system for real-time sorting experiments. As can be seen in Figure 12, the detection efficiency of the Fast R-CNN algorithm was relatively low and was unable to achieve the real-time sorting of fluorescent seeds in actual production. Thus, the YOLOv3, YOLOv5s, and improved YOLOv5s models were used in the classification system in these experiments to compare the detection results of the models.

In the first experiment, to eliminate the influence of mechanical or electrical faults in the sorting system, the actions of the sorting system were manually controlled based on the detection results of the model during the experiments to ensure that the kernels fell into the corresponding boxes. The results of the experiments and the various evaluation indicators are shown in Table 5 and Figure 13.

The proportions of fluorescent and non-fluorescent kernels in the sample were balanced; therefore, accuracy was used as the main indicator for evaluating the performance of the models. Compared with the YOLOv3 algorithm, the target detection efficiency and accuracy of the YOLOv5s algorithm were improved to a certain extent. The improved YOLOv5s model achieved good results due to the gamma transformation for image enhancement and the use of the CBAM lightweight module. The YOLOv3 model had the lowest detection accuracy at just 89.2%, which increased to 91.2% for the YOLOv5s model and 96% for the improved YOLOv5s model. Thus, the improved YOLOv5s model performed the best in the real-time sorting experiment.

In the second experiment, to study the effect of the speed of the sorting system on its accuracy, the improved YOLOv5s model was uniformly used in the sorting system to identify the fluorescent maize kernels. The distance between the seeds while passing through the conveyor was fixed at 30 mm and the speed of the conveyor belt was adjusted for the different seed-dropping speeds to gradually reduce the time between the seeds. The experimental parameters and results are presented in Table 6 (n represents the number of maize kernels) and Figure 14. The actual seed sorting results are shown in Figure 15.

The results show that the accuracy of fluorescent seed sorting decreased with the decrease in the interval between seeds. This is mainly because the increase in the seed-dropping speed increased the probability of dropping two or more seeds at a time from the opening of the seed-drop device, making it difficult to accurately identify and separate the seeds. To further increase the accuracy of the sorting system, the accuracy of the detection algorithm should be increased and the operational reliability of mechanical devices such as the seed-drop device should be improved.

## 4. Conclusions

By addressing the technical requirements of the high-precision real-time sorting of fluorescent maize kernels via MCS technology, an MV system and prototype seed sorting system for identifying fluorescent maize kernels in real time and an improved YOLOv5s model for fast and precise identification of fluorescent maize kernels were developed.

Images of maize kernels with mCherry and DsRed fluorescent proteins and maize kernels without fluorescent proteins were captured by the MV system under different combinations of filters and excitation light sources with varying central wavelengths. An analysis of the RGB and HSV colour feature parameters of the images indicated that using a yellow LED light as the excitation light source and an industrial camera filter with a central wavelength of 645 nm is optimal for identifying fluorescent maize kernels. A comparison of the identification and sorting results of the improved YOLOv5s model developed in this study with those of the classical YOLOv5s and YOLOv3 models revealed that the YOLOv5s network has considerable advantages over the YOLOv3 network regarding various indicators. However, compared to the classical YOLOv5s model, the inference time of the improved YOLOv5s model was slightly longer. However, the accuracy of real-time sorting was significantly increased (from 91.2% to 96%), thereby meeting the requirements of high-precision sorting. Additionally, in real-time sorting experiments, increasing the operating speed of the sorting device increased the probability of dropping two or more seeds at a time from the opening of the seed-drop device, which reduced the sorting accuracy of the device.

The proposed sorting system and the deep learning identification model offer a feasible solution for the high-precision real-time sorting of fluorescent maize kernels, as well as robust support for the industrial application of MCS technology to corn. Furthermore, this study has broad technical value for the efficient identification and sorting of various fluorescently labelled plant seeds. In future research, we intend to further demonstrate and design a more energy-saving and efficient seed-sorting system based on the MV system proposed in this paper by taking into account wind and gravity. Regarding the deep neural network, a pruning algorithm can be used to further optimize the existing YOLOV5 model, as well as reduce its size and inference time, thereby improving the recognition speed of fluorescent seeds.

## Figures and Tables

**Figure 1 sensors-23-02840-f001:**
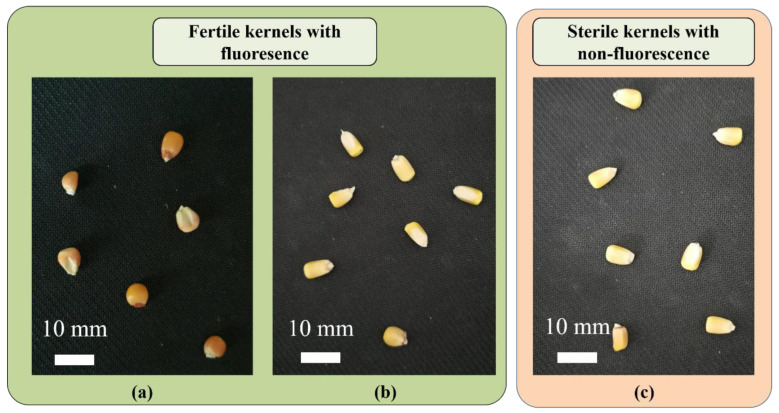
Three types of maize kernels based on the type of fluorescent protein: (**a**) mCherry (mCherry-1), (**b**) DsRed (DsRed-2), and (**c**) no fluorescent protein (Non-3).

**Figure 2 sensors-23-02840-f002:**
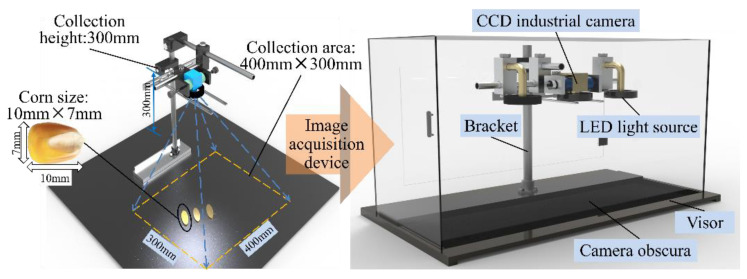
Pictorial representation of the hardware of the machine vision system. CCD: charge-coupled device.

**Figure 3 sensors-23-02840-f003:**
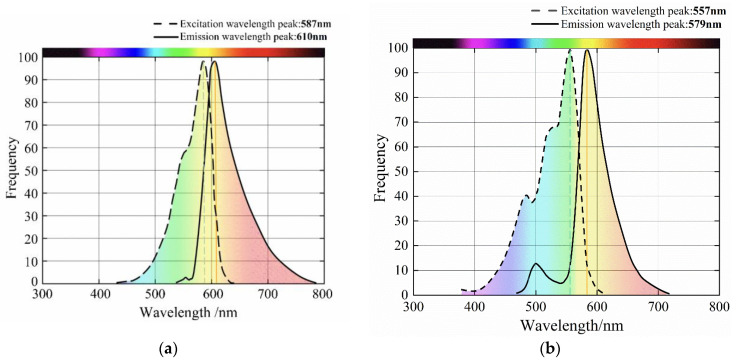
Fluorescent protein spectra: (**a**) mCherry, and (**b**) DsRed.

**Figure 4 sensors-23-02840-f004:**
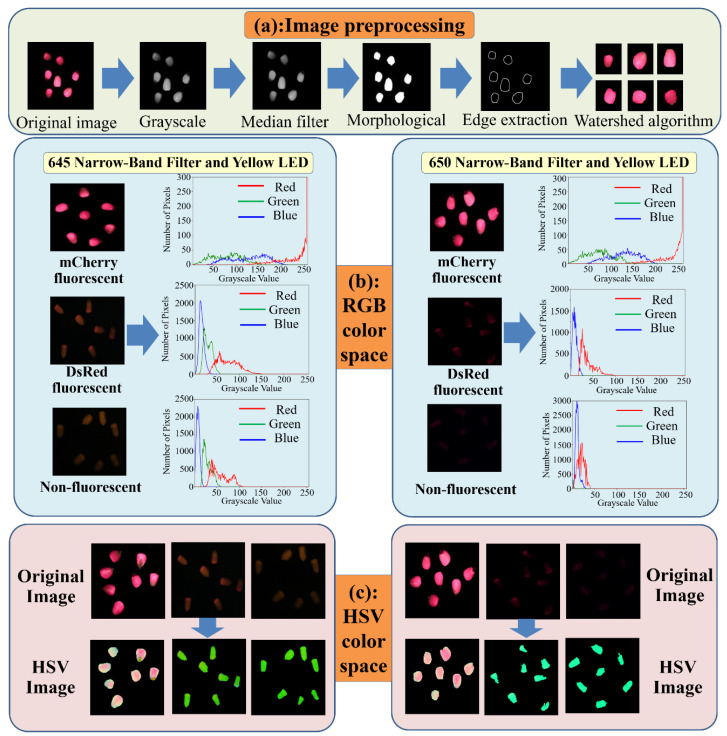
Comparative analysis of excitation light source and filter configurations for fluorescent maize kernels in terms of (**a**) image processing, (**b**) RGB colour space, and (**c**) HSV colour space.

**Figure 5 sensors-23-02840-f005:**
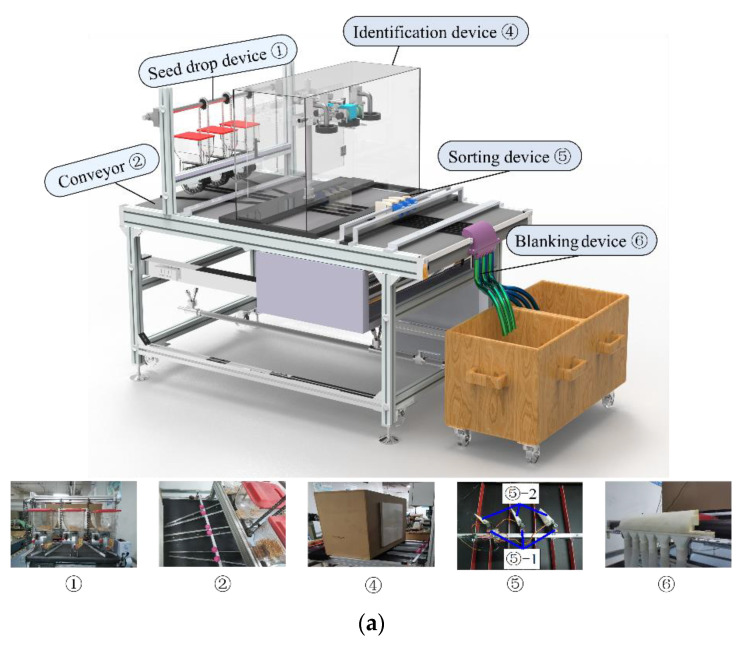
Maize kernel sorting system: (**a**) structural design, and (**b**) control system design.

**Figure 6 sensors-23-02840-f006:**
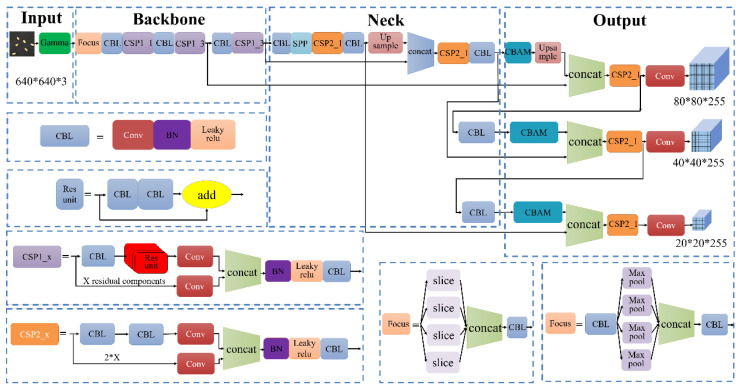
Architecture of YOLOv5s network.

**Figure 7 sensors-23-02840-f007:**
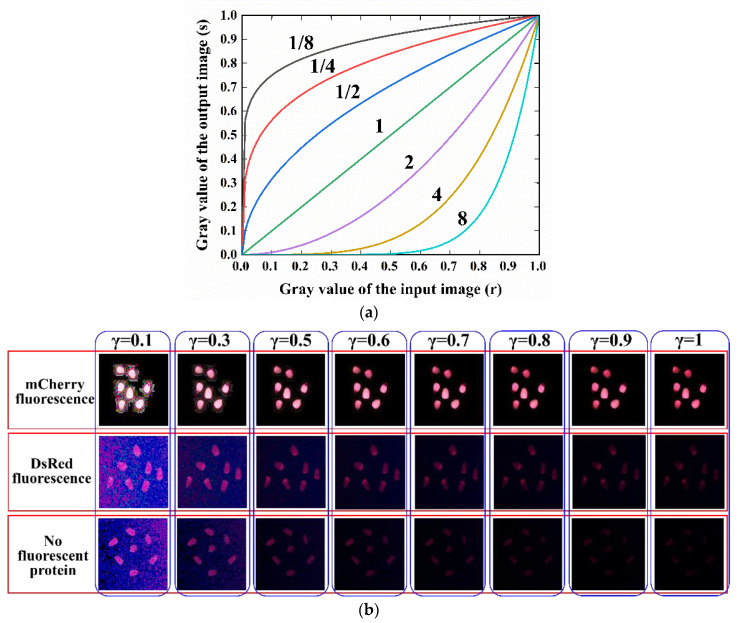
Maize kernel image enhancement with gamma transformation: (**a**) gamma transformation image grey-scale input–output relationship, and (**b**) results of gamma transformation.

**Figure 8 sensors-23-02840-f008:**
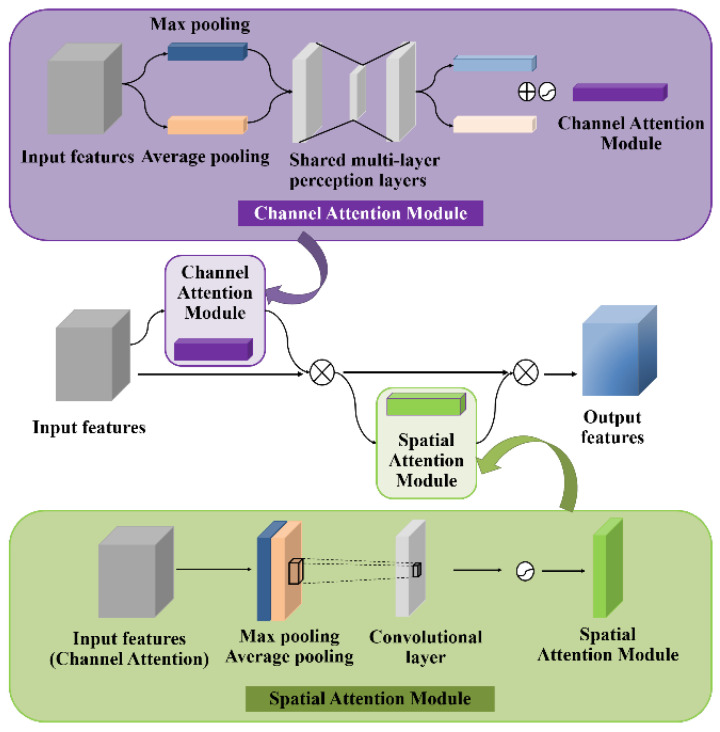
Convolutional block attention module.

**Figure 9 sensors-23-02840-f009:**
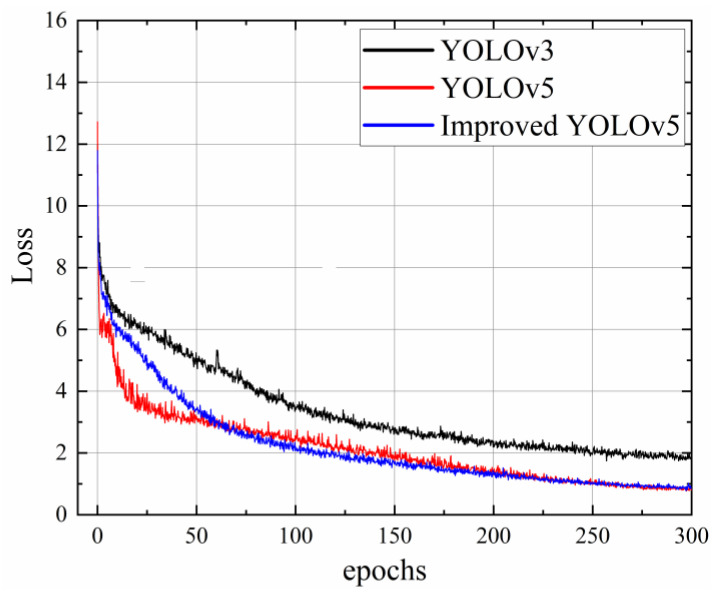
Comparison of loss functions of YOLOv3, YOLOv5s, and improved YOLOv5s models.

**Figure 10 sensors-23-02840-f010:**
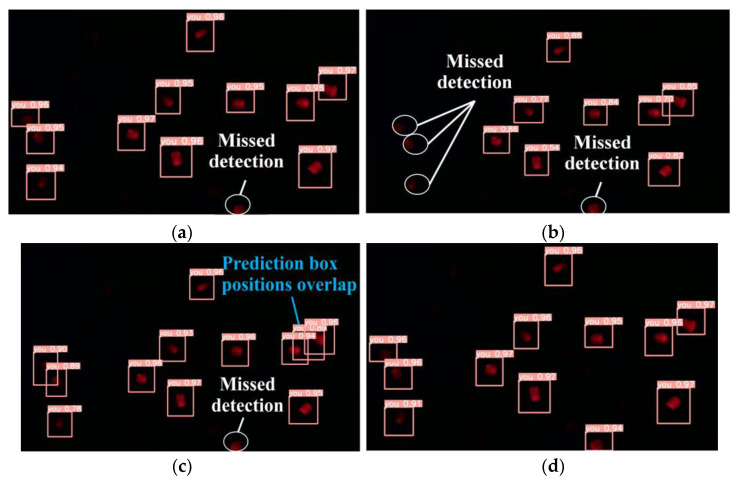
Detection results of the different models: (**a**) Fast R-CNN, (**b**) YOLOv3, (**c**) YOLOv5s, and (**d**) improved YOLOv5s models.

**Figure 11 sensors-23-02840-f011:**
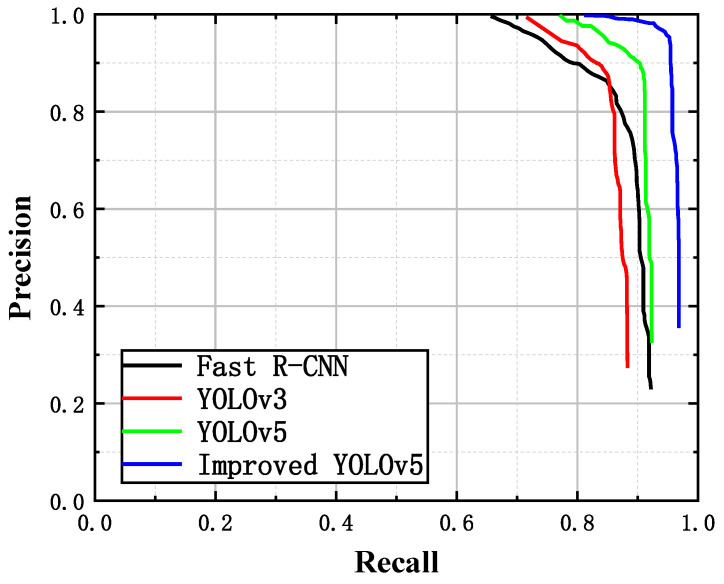
P–R curves for the detection models.

**Figure 12 sensors-23-02840-f012:**
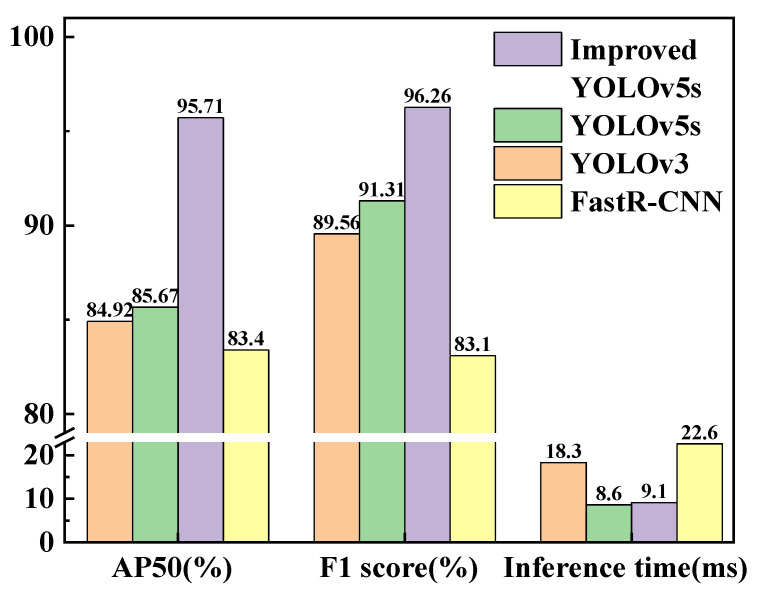
Comparison of evaluation indicators of YOLOv3, YOLOv5s, and improved YOLOv5s models.

**Figure 13 sensors-23-02840-f013:**
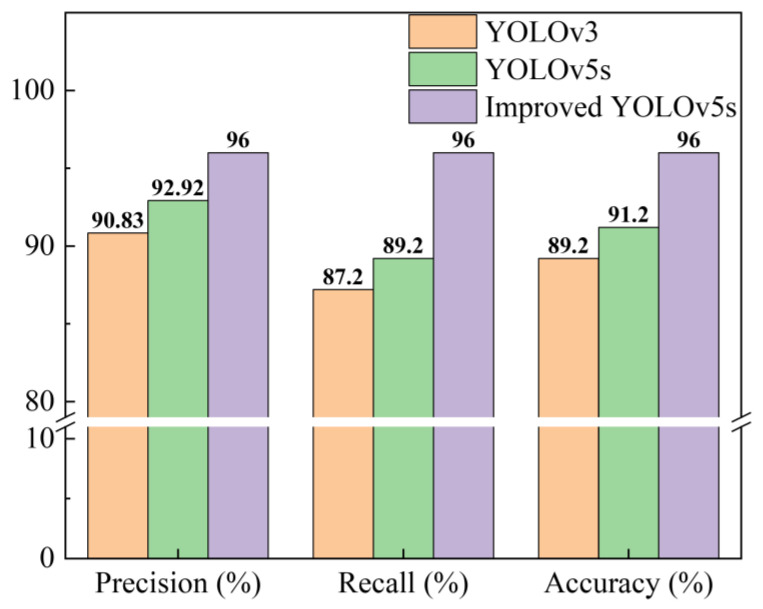
Comparison of sorting results of YOLOv3, YOLOv5s, and improved YOLOv5s models.

**Figure 14 sensors-23-02840-f014:**
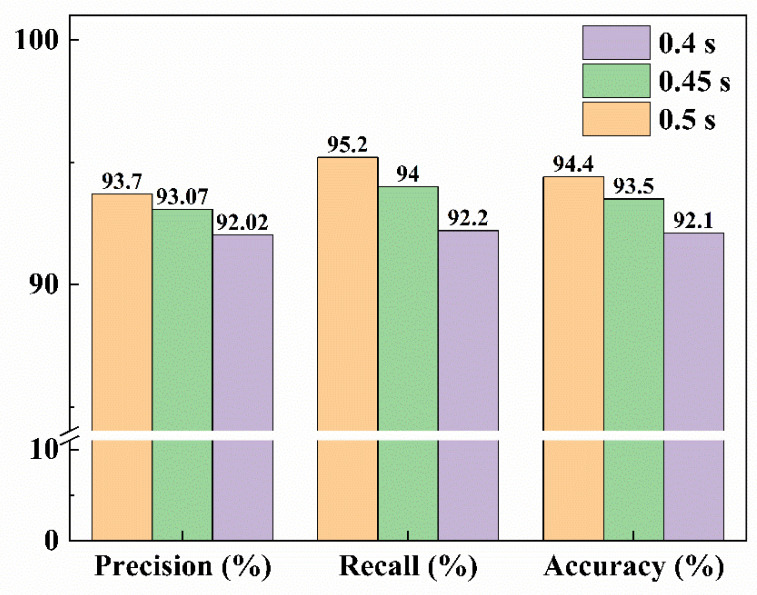
Relationship between speed and accuracy.

**Figure 15 sensors-23-02840-f015:**
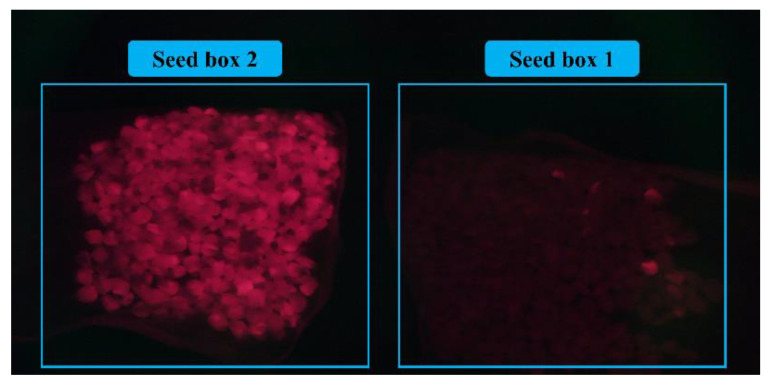
Sorting results of sorting system at kernel intervals of 0.5 s.

**Table 1 sensors-23-02840-t001:** Categories and types of maize kernel samples.

Category	Type	Identifier
Fertile line fluorescent kernels	mCherry fluorescent proteins	mCherry-1
	DsRed fluorescent proteins	DsRed-2
Sterile line non-fluorescent kernels	No fluorescent proteins	Non-3

**Table 2 sensors-23-02840-t002:** Excitation light source and filter configuration scheme for fluorescent maize kernels.

Excitation Light	Filter Central Wavelength	Identifier
Yellow LED light source	645 nm	Yellow+645
Yellow LED light source	650 nm	Yellow+650

**Table 3 sensors-23-02840-t003:** RGB and HSV colour characteristics of the various types of maize kernels.

Light Source Combination	645 nm Narrow-Band Filter and Yellow LED			650 nm Narrow-Band Filter and Yellow LED		
Colour Moment	mCherry Fluorescent	Dsred Fluorescent	Non -Fluorescent	mCherry Fluorescent	DsRed Fluorescent	Non -Fluorescent
R mean	2.9193	0.3407	0.2607	3.3229	0.9551	0.6344
G mean	0.9948	0.0070	0.0008	1.0079	0.3876	0.3271
B mean	1.6094	0.0931	0.1490	1.7842	0.2043	0.0909
R second moment	26.04787	3.9312	2.4627	28.0582	8.8454	6.3863
G second moment	39.2998	0.0075	0.0532	9.1578	3.5720	3.2688
B second moment	60.1261	1.0997	1.4021	15.4567	1.9155	0.9588
R third moment	117.3413	9.3319	5.3579	57.0579	19.0368	14.1845
G third moment	39.2998	0.0075	0.0532	9.1578	3.5720	3.2687
B third moment	73.0132	2.6273	3.0805	32.1373	4.1688	2.1865
H mean	1.7845	0.1232	0.0043	1.4268	0.2006	0.0909
S mean	1.0079	0.0009	0.0002	0.8058	0.3839	0.3271
V mean	3.3229	0.2473	0.0019	2.8280	0.9426	0.6344
H second moment	15.4567	1.1981	0.2997	13.2123	1.8943	0.9587
S second moment	9.1578	0.0219	0.0189	7.8973	3.5609	3.2687
V second moment	28.0582	2.4308	0.1334	25.0905	8.7925	6.3862
H third moment	32.1373	2.6146	1.2391	28.5106	4.1333	2.1864
S third moment	15.7378	0.1438	0.1438	17.7489	7.6404	7.2276
V third moment	57.0579	5.3356	0.5823	52.1457	18.9703	14.1846

**Table 4 sensors-23-02840-t004:** Comparison of evaluation indicators of various models.

Algorithm	AP50	F1 Score	Inference Time
Fast R-CNN	85.41%	86.15%	22.6 ms
YOLOv3	84.92%	89.56%	18.3 ms
YOLOv5s	85.67%	91.31%	8.6 ms
Improved YOLOv5s	95.71%	96.26%	9.1 ms

**Table 5 sensors-23-02840-t005:** Comparison of sorting results of various models.

Algorithm	TP	TN	FP	FN	Precision	Recall	Accuracy
YOLOv3	436	456	44	64	90.83%	87.20%	89.20%
YOLOv5s	446	466	34	54	92.92%	89.20%	91.20%
Improved YOLOv5s	480	480	20	20	96.00%	96.00%	96.00%

**Table 6 sensors-23-02840-t006:** Results of the real-time sorting experiment.

Algorithm	Kernel Interval Time	Kernel Drop Speed (n/min)	Conveyor Belt Speed (m/s)	TP	TN	FP	FN	Precision	Recall	Accuracy
Improved YOLOv5s	0.5 s	360	0.060	476	468	32	24	93.70%	95.20%	94.40%
	0.45 s	405	0.067	470	465	35	30	93.07%	94.00%	93.50%
	0.4 s	450	0.075	461	460	40	39	92.02%	92.20%	92.10%

## Data Availability

Data is not available due to privacy or ethical restrictions.

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
