# Peer review of "Classification of Fluorescently Labelled Maize Kernels Using Convolutional Neural Networks"

_sensors, 2023, doi:10.3390/s23052840_

Round 1

Reviewer 1 Report

The manuscript is written with clear understanding of the project addressed. However, there are minor concerns that need to be addressed to enhance the quality of the manuscript. My specific comments are as follows:

1.     Please double-check the following issues that appear in the manuscript: the format of insertion of all references, the clarity of images, and the regularity of punctuation.

2.     Please re-describe L110 to L117, the current formulation is not suitable to appear in a published paper.

3.     yolov5 includes yolov5s,yolov5m and yolov5l. Please specify which one is chosen in the manuscript.

4.     It is suggested that other comparisons be added to 3.1, not just loss.

5.     The author should not just describe the results, but should argue/explain the findings. The discussion section is missing from the manuscript.

6.     In the authors' description, only the results of training and testing are shown, but the authors are not seen to validate the final model.

7.     It is suggested to add comparison experiments with other CNN models in results. Why just choose yolov3 as the comparison model instead of other models in the yolo series?

8.     Suggest adding some references to support your results.

Reviewer 2 Report

This manuscript, "Classification of fluorescently labelled maize kernels using convolutional neural network" (sensors-2220520), presents a machine vision system developed for real-time classification of fluorescent maize kernels using the YOLOv5 Convolutional Neural Network. The results showed that using a yellow LED light source with a 645 nm central wavelength filter achieved the best recognition effect with 96% accuracy. The study provides a high-precision, real-time solution for classifying fluorescent maize kernels and could be useful for identifying other fluorescently labelled plant seeds.

The introduction, material and methods, results, and discussion sections are interesting, but the material and methods section is dense and needs reformulation for clarity. Please also check all references for consistency with "Instructions for Authors." There were instances of Chinese characters in the manuscript which are not appropriate.

The images and schematics are of good quality and visually interesting. Some suggestions include:

-Putting keywords in alphabetical order;

-Standardizing equipment/reagents/software nomenclature with fabricant, city, state, country (three-letter);

-Checking all manuscript for proper standardization.

Some questions to consider include:

-How does the improved YOLOv5 model compare with the classical YOLOv5 and YOLOv3 models in terms of accuracy and inference time?

-What is the broader technical value of the study for efficiently identifying and sorting various fluorescently labelled plant seeds? Is it possible to apply this method to other species of seeds?

-What is the impact of increasing the sorting device's operating speed on sorting accuracy?

For the abstract, consider the following changes for clarity:

-Change "will achieve the best recognition effect of fluorescent corn seeds" to "will achieve the best recognition effect for fluorescent maize kernels."

-Change "greatly raised the recognition accuracy of fluorescent kernels to 96%" to "greatly increased the recognition accuracy of fluorescent maize kernels to 96%."

For the manuscript:

-In Line 15, what is "A"?

-In Line 82, what is "dim"?

-Lines 107-190 seem unnecessary.

-Lines 110-117 should be reformulated and may not be necessary in the material and methods section.

-In Figure 9 legends (Line 384), what is the circle and what are the units on the y-axis?

-In Lines 408 and 443, what is the y-axis and what are its units? Please add y-axis information.

-Check old references for accuracy.

Best regards

Reviewer 3 Report

This manuscript put forward a deep learning-based approach for classifying fluorescently labelled maize kernels, where convolutional neural network (CNN) was employed for the task of interest. To achieve this target, a fluorescent protein excitation light source was combined with a filter to establish a machine vision system. Finally, the experiments were conducted to validate the performance of the proposed method, with satisfactory results. Overall, the topic of this research is interesting, and the structure of manuscript was well organised. The detailed comments are provided as follows.

1.       The contributions and innovation of the paper should be clearly clarified in abstract and introduction.

2.       Broaden and update literature review on CNN/deep networks and its application. E.g. Vision-based concrete crack detection using a hybrid framework considering noise effect; Torsional capacity evaluation of RC beams using an improved bird swarm algorithm optimised 2D convolutional neural network.

3.       The performance of the CNN model is heavily dependent on the setting of hyperparameters. How did the authors set the network parameters to achieve the best classification accuracy?

4.       More evaluation metrics should be considered for model performance evaluation.

5.       Superiority of the proposed method should be verified via the comparison with other deep learning methods.

6.       More future research should be included in conclusion part.

Round 2

Reviewer 2 Report

Dear, I consider the authors made important changes in the manuscript and it was highly improved.  Best Regards